# Zebrafish and Medaka: Important Animal Models for Human Neurodegenerative Diseases

**DOI:** 10.3390/ijms221910766

**Published:** 2021-10-05

**Authors:** Jing Wang, Hong Cao

**Affiliations:** 1State Key Laboratory of Freshwater Ecology and Biotechnology, Institute of Hydrobiology, Chinese Academy of Sciences, Donghu South Road 7#, Wuhan 430072, China; wangjing@ihb.ac.cn; 2College of Advanced Agriculture Sciences, University of Chinese Academy of Sciences, Beijing 100049, China

**Keywords:** zebrafish, medaka, disease models, neurodegenerative

## Abstract

Animal models of human neurodegenerative disease have been investigated for several decades. In recent years, zebrafish (*Danio rerio*) and medaka (*Oryzias latipes*) have become popular in pathogenic and therapeutic studies about human neurodegenerative diseases due to their small size, the optical clarity of embryos, their fast development, and their suitability to large-scale therapeutic screening. Following the emergence of a new generation of molecular biological technologies such as reverse and forward genetics, morpholino, transgenesis, and gene knockout, many human neurodegenerative disease models, such as Parkinson’s, Huntington’s, and Alzheimer’s, were constructed in zebrafish and medaka. These studies proved that zebrafish and medaka genes are functionally conserved in relation to their human homologues, so they exhibit similar neurodegenerative phenotypes to human beings. Therefore, fish are a suitable model for the investigation of pathologic mechanisms of neurodegenerative diseases and for the large-scale screening of drugs for potential therapy. In this review, we summarize the studies in modelling human neurodegenerative diseases in zebrafish and medaka in recent years.

## 1. Introduction

Neurodegenerative diseases are a major threat to human health. With the increase in the elderly population, these age-dependent diseases are becoming increasingly prevalent [1]. These disorders are devastating to families, and they represent a huge burden for society. Hence, it is urgent to develop novel and more effective therapeutic strategies to remedy these diseases. Animal models were confirmed as a useful tool to investigate the complex mechanisms of neurodegenerative diseases.

Over the past several decades, animal models, such as mice, monkeys, dogs, pigs, fruit flies, and fish, have contributed greatly to our understanding of the genetic basis of the cellular and molecular mechanisms behind neurodegenerative diseases [2,3,4,5,6]. In particular, small fish such as zebrafish (*Danio rerio*) and medaka (*Oryzias latipes*) offer several advantages as model organisms for human neurodegenerative disease studies and drug discovery. Due to their relatively small size and short lifespan, they require less space and are more cost-efficient for laboratory maintenance compared with other vertebrate model organisms, such as the mouse. In addition, they have very high fecundity, and their embryos are transparent during development, which facilitates the non-invasive visualization of its development, and complex mechanisms of neurodegeneration can be analysed more rapidly than in mouse and other vertebrate animal models [7,8,9,10,11].

Finally, drugs can be administered by intraperitoneal injection or oral gavage in adult zebrafish [12] or medaka [13], whereas in larvae or embryos, they are always administered by adding them to the water and drug solution [14]. Due to their small size, they can be easily treated in the 24-well plate, 96-well plate, or 10-cm Petri dish. This facilitates subsequent analysis of phenotypes after drug treatment. Therefore, all these characteristics make them suitable for large-scale and high-throughput drug screening scans. 

On the other hand, the identity of nucleotide or amino acid sequences between zebrafish and human homologues is approximately 71% [15], which is much higher than some invertebrate animal models such as roundworms (*Caenorhabditis elegans*) (30–60%) [16] and fruit flies (*Drosophila melanogaster*) (40%) [17]. Notably, zebrafish possess a vertebrate neural structural organisation, and all of the major structures are similar to the mammalian brain. Furthermore, zebrafish also possesses a functional Blood–Brain Barrier (BBB), similar to humans [18]. Many important neurotransmitters were detected in the neurotransmitter profile of zebrafish, which is very important for neuroscientific studies [19].

Although the zebrafish is the most widely used fish model globally, medaka is also used extensively, especially in Europe and Asia [20]. Compared with the zebrafish, the embryos of medaka tolerate a wider temperature range (4–35 °C until the onset of heartbeat and 18–35 °C thereafter, compared to 25–33 °C in zebrafish) [11,21]. This provides great convenience in screens for isolation of low temperature-sensitive gene mutations and the manipulation of developmental rates [11]. In addition, medaka has a long history as a genetic model system. Therefore, a lot of inbred strains from different populations with a high degree of genetic polymorphism are available. This facilitates the generation of high-resolution genetic maps and the genetic analysis of monogenic traits and quantitative trait loci [21].

Therefore, all these factors make zebrafish and medaka of great value in studies of neurodegenerative diseases [22]. As a result, the publications in PubMed using zebrafish, the more popular model of the two, as the neurodegenerative disease model increased sharply in recent years (Figure 1). This review summarizes the use of zebrafish and medaka as models in neurodegenerative disease research.

## 2. Parkinson’s Disease Models

Parkinson’s disease (PD) is one of the most common neurodegenerative diseases that affects the motor system. Surveys, medical records, and death certificates demonstrate that the prevalence of PD has notably increased worldwide in recent years, possibly due to the growing elderly population worldwide [23,24,25]. The prevalence of PD was approximately 8.52 million and the incidence was 1.02 million in 2017 globally [26], whereas approximately 0.34 million people died from PD in 2017 globally [27]. It is predicted that the number of cases will reach 12 million by 2050 [28]. In spite of extensive studies that focus on the epidemiology and possible treatments of PD, its pathogenic mechanism has not been fully elucidated, and there is still no effective therapeutic strategy to cure this disease [29]. Compared with some traditional mammal models such as mice, zebrafish and medaka have comparative advantages for the pathological research of PD due to their short life cycles and high fecundity, which makes them particularly suitable for large scale drug screening [14,30,31,32]. In addition, as vertebrate species, zebrafish and medaka have higher genetic similarity to humans than invertebrate model animals such as roundworms and fruit flies [15,16,17,20]. In this review, we summarize several studies of PD in zebrafish, focusing on those published in recent years (Table 1) and several studies of PD in medaka. We discuss two main types of models: neurotoxin-induced and genetic models.

### 2.1. Neurotoxin-Induced PD Models in Zebrafish and Medaka

Some neurotoxins, such as 1-methyl-4-phenyl-1,2,3,6-tetrahydropyridine (MPTP), 6-hydroxydopamine (6-OHDA), paraquat (1,1’-dimethyl-4,4’-bipyridinium dichloride), and rotenone are often used to selectively induce harmful effects to the dopaminergic neurons, which leads to dopaminergic neuronal loss and increases the risk of PD in model animals.

#### 2.1.1. MPTP

MPTP is a neurotoxin that is widely used in the animal models of PD. It targets dopaminergic neurons and induces their loss in the substantia nigra, which leads to the appearance of motor features of PD [66,67,68]. It was observed that MPTP can induce PD-like symptoms in other animal species such as rodents, primates, and zebrafish [9]. A recent study demonstrated that the mechanism of action of MPTP toxicity is conserved both in zebrafish and humans [33], which suggests that it may be suitable to be used as a neurotoxin to induce the PD model in zebrafish. After the exposure to MPTP, zebrafish exhibited obvious movement impairment, including reduced locomotory activity and aberrant swimming behaviour, equivalent to the bradykinesia-like symptoms in humans. In addition, the exposure to MPTP also weakens the tactile sensitivity in zebrafish [34,35]. Another study found that the intraperitoneal injection of MPTP in adult zebrafish caused a significant reduction of locomotory activity, accompanied by the loss of dopaminergic neurons and the over-expression of synuclein in the optic tectum [36]. Furthermore, 78 proteins differentially expressed in the brains of MPTP-treated zebrafish (compared to the control) were involved in several neurological pathways [36].

Similarly, the MPTP also effectively triggered PD-like symptoms in medaka [69]. Shortly after the MPTP-exposure treatment, an obvious reduction of swimming movement was observed in the larvae of medaka. The histochemical examination of the medaka brain showed that the immunoreactivity of tyrosine hydroxylase (TH)-positive neurons in the diencephalon almost totally disappeared, suggesting a damage to the dopamine system [69,70].

#### 2.1.2. 6-OHDA

6-OHDA is a synthetic neurotoxin which also has been used to construct the PD animal models. In zebrafish, after the injection of 6-OHDA into the ventral diencephalon, a significant reduction of dopaminergic neurons and reduced swimming speed and distance travelled were observed [37]. A recent study also reported the death of DA neurons and the reduction of dopamine levels in the lesioned brain area of the 6-OHDA administrated zebrafish [38]. Similarly, the exposure of zebrafish larvae to 6-OHDA produced a reduction in dopaminergic neurons [39]. Morphological alternations were also observed in the 6-OHDA-exposed zebrafish embryos (blood pooling, cardiac edema, delayed development) [40] and larvae (reduced total length and head length) [39].

Interestingly, there was no substantial reduction in the dopaminergic neurons in the medaka after 6-OHDA exposure [70]. The authors suspected that 6-OHDA could not permeate to the BBB in the adult fish, so they injected 6-OHDA directly into the cerebrospinal fluid (CSF) of medaka, which did cause a loss of dopamine neurons. Similar to the MPTP treatment in medaka, only the middle diencephalic cluster among the dopamine neuron clusters exhibited a robust cell loss [70].

#### 2.1.3. Paraquat

Paraquat is a herbicide that has similar structure to the MPTP [71]. Its effects on zebrafish have been tested using embryonic, larval, and adult stages. The exposure of zebrafish embryos to paraquat triggered premature hatching and increased locomotory activity in the larvae [41]. Another study showed that the exposure of 18 hpf (hours post-fertilization) embryonic zebrafish to a low dose of paraquat (0.04 ppm) produced neurodegenerative phenotypes and motor deficits in 50% of the embryos. Furthermore, various developmental anomalies including small eyes, flat head, a pinched midbrain–hindbrain boundary, thin yolk extension, and curved-up body were detected in the hatched larvae [42]. However, in contrast to the observations in mice models, the exposure to paraquat resulted in an increased dopamine level in adult zebrafish, which indicated that the paraquat-treated zebrafish did not recapitulate the PD pathology [43]. Moreover, further studies showed that both the expression level of TH and the numbers of dopaminergic neurons were not affected in the paraquat-treated zebrafish [41,44]. Nevertheless, the paraquat-treated zebrafish still showed impaired locomotory activity [43,44,72].

#### 2.1.4. Rotenone

Rotenone is a rotenoid extracted from the roots of plants from the Leguminosae family, commonly used as an insecticide, herbicide, and piscicide [73]. It can easily permeate the BBB and enter the central nervous system (CNS) [73,74], and its destructive effect on dopaminergic neurons in rodents makes it a popular drug of choice to trigger PD models [75,76]. In zebrafish, rotenone also reduces the dopamine population, marked by a decreased expression level of TH [45,46]. Studies also showed that after the rotenone administration, adult zebrafish exhibited reduced swimming speed and shorter travel distances compared to the control group [46,47]. As mentioned above, this aberrant swimming behaviour is reflective of bradykinesia-like symptoms of PD in humans [45]. In addition, a study found impaired olfaction in a rotenone-administrated zebrafish, which is a typical non-motor symptom of PD [48].

### 2.2. Genetic PD Models in Zebrafish and Medaka

Accompanied by the development of gene-editing technology, extensive studies have applied genetic PD models to study the PD development in zebrafish and medaka. These widely used methods include the MO (morpholino antisense technology) [77], TALEN (transcription activator-like effector nucleases technology) [78] and CRISPR/Cas9 (clustered regularly interspaced short palindromic repeats/Cas9 technology) [79].

#### 2.2.1. Parkinson’s Disease Protein 2 (PARK2)

Mutations of the *PARK2* gene are correlated with the early onset of PD, and they are the most prevalent cause of autosomal recessive PD [80]. *PARK2* encodes the Parkin protein, a ubiquitin ligase that targets other proteins for degradation. A loss-of-function Parkin protein encoded by a mutated *PARK2* gene will destroy its function, which in turn increases the risk of inducing PD [49]. The zebrafish Parkin protein consists of 458 amino acids and exhibits 62% identity to the human orthologue, with 78% identity in functionally relevant regions [50]. In addition, although neurodegenerative diseases are always late onset in human patients, we can still detect some similar pathological characteristics and phenotypes in the early developmental stages of zebrafish, therefore, the administration of MO technology could reveal some important mechanisms of neurodegenerative diseases. For instance, a previous study using the MO-mediated knockdown of *PARK2* in zebrafish reported that knockdown of *PAKR2* gene in zebrafish displayed a 20% loss of the DA neuron numbers in the diencephalon, but the swimming behaviour was not apparently affected [50]. However, the MO technology still has some disadvantages, such as the occurrence of false positives, cytotoxicity, and the inability to obtain stable genetic lines. The negative phenotype of Parkin-mutant medaka was also similar to Parkin deficient mice [70,81].

#### 2.2.2. PTEN (Phosphatase/Tensin Homolog)-Induced Kinase 1 (Pink1)

The PINK1 protein is the second most common cause of autosomal recessive early onset cases in PD patients [49,82,83]. The zebrafish *Pink1* gene shares 57.8% identity to the human orthologue [51]. A previous study using the antisense MOs injection into the zebrafish embryo showed a reduction of dopaminergic neurons, but the locomotory activity remained unchanged [51]. However, another study of MO of *Pink1* in the zebrafish embryos did not detect a loss of dopaminergic neurons, but obviously disorganized diencephalic dopaminergic neurons were observed [52]. It is possible that these differences may have been caused by different target regions of this gene in different studies, which remains to be elucidated in the future studies.

Some studies used the Targeting Induced Local Lesions In Genomes (TILLING) library [70] to screen for loss-of-function mutations in human autosomal recessive PD homologous genes in medaka and identified a mutation in the homologous medaka *Pink1* gene, the *Pink1*^Q178X^ mutation [70]. RT-PCR and in situ hybridization analysis detected a relatively small number of PINK1 transcripts, demonstrating its loss-of-function at the mRNA level. A homozygous *Pink1*^Q178X^ medaka showed some phenotypes similar to the human PD patients, such as late-onset reduction of spontaneous movements and a decreased level of 3,4-dihydroxyphenylacetic acid, a dopamine metabolic product [70].

#### 2.2.3. LRRK2 (Leucine-Rich Repeat Kinase 2)

Mutations in the *LRRK2* gene are the most prevalent underlying reason for the late-onset autosomal dominant PD [54]. The LRRK2 protein consists of several repeat regions followed by ROC (Ras of complex proteins)-COR (C-terminal of ROC) GTPase, kinase, and WD40 domains [53]. A previous study using the MO-mediated *LRRK2* knockdown in zebrafish displayed a severe embryonic lethality, and the surviving larvae morphants showed several developmental aberrations, including slow growth, heart edema, reduced brain size, and reduced number of dopaminergic neurons [54]. Another study employing the deletion of the *LRRK2* WD40 domain revealed diencephalic DA neurodegeneration and a shortened swimming distance in zebrafish [55]. Some recent studies observed that mutations of the *LRRK2* gene caused a loss of neuronal cells and synuclein aggregation in zebrafish, similar to the PD phenotype in humans [56].

#### 2.2.4. PARK7 (Parkinson’s Disease Protein 7)

The *PARK7* gene encodes a protein named DJ-1, which plays an important role in the antioxidative stress reaction and the protection of neuron survival [84,85]. It was reported that mutations in the *PARK7* could induce the early onset of PD [57]. The zebrafish *PARK7* gene encodes a 189 amino acid protein, which share 83% identity to the DJ-1 of humans [58]. A previous study showed that a transient inactivation of *PARK7* mediated by the MO-injection in zebrafish embryo inactivated the DJ-1 production, and the knockdown of *PARK7* caused dopaminergic neuronal cell death without toxin exposure [59]. In a recent study, *PARK7*-knockout zebrafish line was constructed using the CRISPR/Cas9 method, by targeting the exon 1 of *PARK7*. This DJ-1 deficient zebrafish appeared to develop normally during the larval and young adult stages. With aging, however, they exhibited lower TH levels, respiratory failure in skeletal muscle, and lower body mass, particularly pronounced in male fish [60].

#### 2.2.5. Other Genes

The *SNCA* gene in humans encodes the α-synuclein protein. It was reported that misfolded α-synuclein proteins form aggregates and thus cause the formation of Lewy bodies (LBs) [86], which is a hallmark of PD pathology [87]. However, the zebrafish lacks the α-synuclein; instead, it possesses three synuclein genes named *SNCB*, *SNCG1*, and *SNCG2*, which encode for β-, γ1-, and γ2-synuclein proteins, respectively [88]. Some previous studies constructed transgenic models that express human wild-type α-synuclein protein in zebrafish [61,89,90]. It was observed that the overexpression of α-synuclein protein in the zebrafish PD model caused a moderate cell death in larval zebrafish sensory neurons [61]. Another study showed that the overexpression of α-Synuclein in zebrafish is a critical factor of the mitochondrial dysfunction in dopaminergic neurons [90].

Mutations in the *GBA* gene are very important causes of PD [91]. In a recent study, a *GBA^−/−^* mutant zebrafish line was generated through the TALEN-mediated technology [62,63]. These *GBA*-null homozygous fish showed a decreased level and activity of the GBA protein in the brain, and reduced numbers of dopaminergic and noradrenergic neurons were detected in the 3-month-old mutants [63].

## 3. Alzheimer’s Disease Models

Alzheimer’s disease (AD) is a neurodegenerative disease characterized by progressive memory loss, cognitive impairment, behavioural changes, and loss of functional abilities [92,93,94,95]. AD is the most prevalent form of dementia. It is estimated that nowadays more than 50 million people worldwide have dementia, and this number is expected to reach over 150 million by the 2050 [93]. AD is irreversible and it causes about 70% of all dementia cases [94]. Unfortunately, it still cannot be prevented, treated, or cured. The drug discovery for AD is very challenging, so no new drugs have been approved since 2003, when Memantine was approved [92,93,94,96]. Including four drugs that are approved by the Food and Drug Administration (FDA), at present there are only five approved drugs on the market available for the treatment of AD [94,97]. In addition, a previous study demonstrated that oxidative stress may induce behavioural and cognitive impairments in the aging zebrafish, just as it does in mammals [98]. Below, we describe previous studies of the molecular mechanisms of AD in zebrafish (Table 2).

### 3.1. Neurotoxic Agents-Induced AD Model in Zebrafish

Although the pathological mechanisms of AD progression remain to be elucidated, two principal neuropathological hallmarks are widely approbated: the extracellular accumulation of the amyloid β (Aβ) plaques and the intracellular neurofibrillary tangles (NFT) [113,114,115]. In addition, several previous studies have reported that the elimination of Aβ through a vascular route is a pivotal mechanism and that its failure could induce the formation of vascular abnormalities, which consequently leads to AD pathogenesis [116,117]. It was reported that Aβ can be degraded in the extracellular space by diverse proteases, such as insulin-degrading enzymes and neprilysin [118]. In addition, cells in the neurovascular unit also have the capacity to endocytose Aβ and eliminate it through lysosomal degradation [119]. In addition, the transportation of Aβ across the BBB also contributes to the removal of soluble interstitial Aβ from the brain [116]. In addition, altered CSF dynamics were widely observed in many AD patients [120]. It might cause an imbalance in the production and clearance of soluble Aβ, thus causing the accumulation of interstitial Aβ, with increased likelihood of plaque formation [121]. When amyloid-β42 (Aβ42) peptides were injected into the cerebral ventricle of adult zebrafish, they aggregated in the brain, where they were mostly seen as intracellular depositions, but also around the blood vessels [99]. Based on this AD model, a subsequent study compared young and old zebrafish to investigate the effects of aging on regenerative ability after Aβ42 deposition [100]. Some previous studies already demonstrated that aging aggravates the neurodegeneration phenotypes and impedes the proliferative response of stem cells in various animal models [122,123]. A study revealed that Aβ42 toxicity causes synapses to degenerate at a higher level in the ventricular region of old zebrafish (1.5-year-old) compared with young adult zebrafish (6 months of age); the same study also found that young zebrafish seemed to activate macrophages and produce newborn neurons at a much higher rate than old zebrafish [100]. The authors of that study speculated that the activated microglia help the zebrafish brain to limit the synaptic degeneration and to promote neurogenesis after Aβ42-induced neurodegeneration [100].

Okadaic acid (OA) is an inhibitor of two main cytosolic, broad-specificity protein phosphatases, PP1 and PP2A. It has been widely used in studies of neurodegeneration in various cell types and rodent models [124]. OA induced AD pathologies in several studies. For instance, it was observed that a decrease in PP2A expression and function caused by OA directly triggered the pathophysiology of AD [101]. In zebrafish, Nada et al. observed some major morphological hallmarks such as mini hemorrhagic transformations or micro-bleeds at the later stages of AD in the OA-induced fish [102]. Another separate research utilized the OA-treated zebrafish for drug screening [103]. In this study, researchers found that LKE (lanthionine ketimine-5-ethyl ester), a derivative of a naturally occurring brain sulfur metabolite, can exhibit a neuroprotective function against OA by improving the levels of the brain-derived neurotrophic factor (BDNF), anti-apoptotic kinase pAkt (Ser473), and the transcription factor phospho-cAMP response element-binding protein (pCREB) (Ser133). When simultaneously exposed to OA and LKE, the zebrafish were able to successfully perform the learning and memory ability, whereas, when exposed to the OA only, the fish could not demonstrate learning and memorizing ability. This suggests that LKE can inhibit the cognitive impairment triggered by OA [103].

Cigarette smoke extract (CSE) consists of 4500 identified and almost 1000 unidentified molecules, many of which could lead to oxidative damage and trigger pro-inflammatory and carcinogenic reactions [125]. Several recent studies revealed that after the CSE treatment, an elevation in the acetylcholinesterase activity was detected in zebrafish, which contributed to its cognitive impairment [104,105].

### 3.2. Metals and AD in Zebrafish

Metals are abundant in our environment. The disturbance of the metal homeostasis is harmful for brain health and can induce dementia. Some studies used metals to trigger memory loss and neurodegeneration in zebrafish. For example, Xu et al. reported that the lead (Pb) exposure caused learning impairments that persisted for at least three generations, showing trans-generational effects of embryonic exposure to Pb [126]. Chronic Aluminum (Al) exposure is regarded as a risk factor for the AD pathogenesis [127,128]. A previous study revealed that after the aluminium chloride (AlCl_3_) treatment, adult zebrafish exhibited AD-like behaviour in locomotory activity tests and activity-avoidance conditioning paradigms, suggesting that locomotory activity and learning and memory abilities were damaged in the Al-exposed zebrafish [106]. Acosta et al. exposed larvae and adult zebrafish to another metal, copper (Cu) [107]. The exposure to the 60 μg/L concentration of Cu produced decreased body length of larvae, spatial memory of adults, and the glutathione S-transferase (GST) activity in the gill [107]. In a recent study, Altenhofen et al. investigated the effects of manganese (Mn) exposure on the cognition and exploratory behavior in larval and adult zebrafish [108]. It caused an impairment of the aversive long-term memory in the MnCl2-treated adult zebrafish and reduced distance traveled and movement time in both larvae and adult fish [108].

### 3.3. Genetic Technology-Mediated AD Model in Zebrafish

Tau is an important microtubule-associated protein for tubule formation which is widely expressed in neurons. Hyperphosphorylation of the tau protein causes the intracellular neurofibrillary tangle (NFT) formation, which is one of the neuropathological hallmarks related to AD [109,110,129]. In a previous study, a transgenic zebrafish line was generated through the Gal4-UAS system. A mutant human gene, *4R/2N-Tau^P301L^*, was expressed under the regulation of a novel bidirectional UAS promoter, which allows the co-expression of a red fluorescent reporter gene in Tau-expressing cells [110]. It was observed that a high level of the mutant P301L Tau protein triggered a transient motor phenotype during the embryogenesis, likely caused by the peripheral motor axonal developmental abnormalities. This result is very important because phenotypic abnormalities at the larval stages make it suitable for high-throughput screening [110].

## 4. Huntington’s Disease Models

Huntington’s disease (HD) is an autosomal dominant, incurable, and fatal neurodegenerative disorder. Initially, HD patients display excessive movements of the limbs and face, and then gradually progress to exaggerated body movements described as chorea. Patients exhibit progressive symptoms, such as psychiatric, cognitive, and motor dysfunction, and this disease is usually lethal 10–20 years after the onset [130,131,132]. HD is caused by an expansion of the polyglutamine-coding region in the N-terminus of the huntingtin protein (HTT) [133]. HTT is a 350 kDa protein that is ubiquitously expressed, evolutionarily conserved, and likely to be involved in many cellular processes [134,135,136]. However, the precise mechanisms underlying the functions of the HTT gene remain incompletely understood.

The zebrafish HTT protein consists of 3121 amino acids and shares 70% identity with the human HTT orthologue [137]. Compared with the HTT-null mutation mice [135], HTT-null mutation zebrafish are viable, so the zebrafish is believed to be a suitable model to study the mechanisms of HD. To investigate the roles of HTT, several previous zebrafish HD models used MO to observe the effects of HTT deficiency in the early zebrafish development [138,139,140]. One study revealed that HTT-deficient zebrafish had hypochromic blood because of the decrease in hemoglobin production, despite the presence of iron within blood cells, and speculated that the disturbance of HTT’s normal function in the iron pathway leads to HD pathology and especially to its neuronal specificity [138]. By use of the same HTT-deficient model, Henshall et al. reported the effects of the loss-of-function of HTT on the developing nervous system and found obvious defects in the morphology of olfactory placode, neuromasts, and branchial arches, which led them to postulate that HTT may have a specific function that enables the formation of telencephalic progenitor cells and preplacodal cells in the forebrain [139]. Another study of the morpholino-based HTT loss-of-function zebrafish observed massive apoptosis of neuronal cells, accompanied by impaired neuronal development, small eyes and heads, and the enlargement of brain ventricles. Interestingly, it was observed that the expression of brain-derived neurotrophic factor (BDNF) was reduced. Notably, treatment of HTT-MO zebrafish embryos with exogenous BDNF rescued these defects, which suggests that increasing the BDNF expression might be a useful strategy for HD treatment [140].

In addition, some scientists established HD zebrafish models through the transgenic technology [141,142,143]. Schiffer et al. transiently expressed 102 polyglutamine repeats in the N-terminal fragment of the HTT protein fused with GFP (Q102-GFP) in zebrafish and found an accumulation of this mutant protein in large SDS-insoluble inclusions in the zebrafish embryos, thus reproducing an important feature of the HD pathology. The expression of the mutant HTT protein resulted in an increase in abnormal morphology and the occurrence of apoptosis in zebrafish embryos. A further study found that soluble mutant HTT protein forms are responsible for toxicity and aberrant polyglutamine aggregates in zebrafish [141]. The same study also found that its toxicity can be suppressed by the heat-shock proteins Hsp40 and Hsp70. Importantly, by the use of this HD transgenic model, two inhibitors of the Q102-GFP aggregation in vivo were identified, both of which are compounds of the *N′*-benzylidene-benzohydrazide class (293G02 and 306H03). In another study, a stable transgenic zebrafish line, which expressed a Q71-GFP fusion protein under the control of the rhodopsin promoter, was constructed to screen FDA-approved drugs to identify novel autophagy-inducing pathways. Three drugs (L-type Ca^2+^ channel antagonists, the K^+^_ATP_ channel opener minoxidil, and the G_i_ signalling activator clonidine), which participate in a cyclical mTOR-independent pathway that regulates autophagy, were detected. This pathway has lots of candidate points to induce autophagy and reduce aggregates [142].

Cre-*loxP* system was also sometimes used to generate conditionally inducible transgenic zebrafish to study HD. For example, Veldman et al. created an inducible zebrafish HD model, in which the N-terminal 17 amino acids (N17) in the context of the exon 1 fragment of HTT were deleted, coupled with 97Q expansion (mHTT-ΔN17-exon1). That study found that, compared with the mHTT-ΔN17-exon1 line, fish with intact N17 and 97Q expansion (mHTT-exon1) had more delayed-onset movement deficits with slower progression. This model confirmed that the deletion of N17 terminal amino acids of the HTT will lead to an accelerated HD-like phenotype in zebrafish [143]. Recently, a separate study treated a transgenic HD zebrafish model with a phosphodiesterase 5 (PDE5) inhibitor and found an obvious decrease in the mutant HTT protein levels, cell death, and morphological abnormalities [144].

## 5. Other Neurodegenerative Disease Models

In addition to the above studies, zebrafish and medaka were also used in the investigation of some other rare neurodegenerative disorders. Amyotrophic lateral sclerosis (ALS) is a neurodegenerative disease characterised by the motor neuron loss, and thus progressive muscle weakness and eventual death, primarily due to respiratory failure. The most prevalent genetic cause of ALS and frontotemporal dementia (FTD) is a hexanucleotide repeat expansion (HRE) within the first intron of the C9orf72 gene [145,146]. Shaw et al. generated two zebrafish lines to express C9orf72 HREs. This model recapitulates the motor deficits, cognitive impairment, muscle atrophy, motor neuron loss, and mortality in early adulthood that was observed in human C9orf72-ALS/FTD patients. Moreover, this stable transgenic model represents a powerful potential for the screening of therapeutic compounds [147]. In another study, several transgenic C9orf72-associated repeat zebrafish lines were generated by TOL2-mediated transposition. These models confirm the poly-GA toxicity in zebrafish. The reduction of poly-GA protein rescues toxicity, indicating its potential therapeutic value to treat C9orf72 repeat expansion carriers [148]. Conversely, Yeh et al. constructed two transient loss-of-function zebrafish larvae (C9orf72^u-DENN^, C9orf72^c-DENN^) using a morpholino injection. These models facilitate advances in the understanding of the functions of C9orf72 and provide potential mechanisms to elucidate the pathogenesis of ALS-FTD [149]. Mutations in the superoxide dismutase 1 (SOD1) gene were identified as another cause of ALS. In a previous study, by outcrossing the G93Ros10-SH1 line with the wildtype AB zebrafish strain, a mutant SOD1 zebrafish model was generated and used for high throughput screening to identify neuroprotective compounds [150].

Spinocerebellar ataxias (SCAs) are global neurodegenerative diseases leading to motor discoordination, which is always caused by the affected cerebellar Purkinje cells (PCs). A previous study generated a transgenic SCA type 13 (SCA13) model, which mimics a human pathological SCA13^R420H^ mutation. This model exhibited neuropathological and behavioural changes similar to those manifested by SCA-affected patients [151]. Based on the same model, Namikawa et al. reported an SCA13-triggered cell-autonomous PC degeneration, which results in eye movement deficits [152]. In a previous study in our lab, we constructed an *NPC1* knock-out zebrafish model using the CRISPR/Cas9-mediated technology [153]. This model developed symptoms similar to those observed in human patients of Niemann-Pick type C disease (NPC). We observed the loss of Purkinje cells in the cerebella of the *NPC1^−/−^* homozygous fish [153] and the aberrant motor behaviour, i.e., ataxias, a typical pathological character of human NPC1 patients (unpublished data), indicating its potential value for investigating the molecular mechanisms of NPC1.

In addition, a previous study generated a Gaucher disease (GD) model in medaka by the use of a high-resolution melting assay in the TILLING library for the *glucocerebrosidase* (*GBA*) gene [154]. In this study, it was observed that the *GBA^W337X/W337X^* (*GBA^−/−^*) medaka displayed complete deficiency in GCase activity, and it showed similar pathological phenotypes with human neuronopathic GD. Importantly, compared with the perinatal death in humans and mice lacking the GCase activity, the *GBA^−/−^* medaka survived for months, enabling the investigation of the pathological progression [154].

## 6. Conclusions

Various zebrafish and medaka models have been generated for neurodegenerative disorder studies in recent years. Compared with some other vertebrate models, such as rodents, these small fish are particularly well-suited for forward and reverse genetic and high-throughput screens for chemical compounds and optical analysis in vivo. These advantages should be considered thoroughly when designing a study aimed at neurodegenerative disorders.

## Figures and Tables

**Figure 1 ijms-22-10766-f001:**
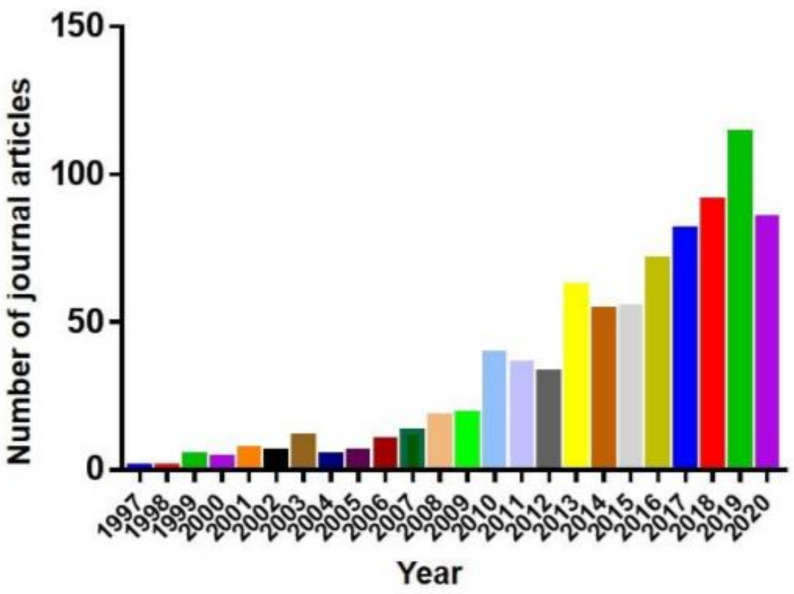
Absolute number of articles for zebrafish neurodegenerative diseases modelper year of publication extracted from the PubMed database.

**Table 1 ijms-22-10766-t001:** Zebrafish models of Parkinson’s disease.

Method	Phenotype	Results	Reference
MPTP induced	Motor impairments and weakened touch sensory	Reduction of locomotor activity and dopaminergic neuron, over-expression of synuclein in the optic tectum	[33,34,35,36]
6-OHDA induced	Motor impairments and anxiety	Reduction of dopaminergic neurons and morphological alternations	[37,38,39,40]
Paraquat induced	Motor impairments, various developmental anomalies	The paraquat-treated zebrafish did not recapitulate PD pathology	[41,42,43,44]
Rotenone induced	Motor impairments, anxiety, and olfactory dysfunction	In addition to motor impairments, they also show Olfactory dysfunction, which is a typical non-motor symptom of PD	[45,46,47,48]
*PARK2* Morpholino	No abnormalities in swimming behavior	Loss of the DA neuron numbers in the diencephalon, whereas no abnormalities in swimming behavior	[49,50]
*PINK1* Morpholino; Transgenes	Motor impairment and oxidative stress	Reduction of dopaminergic neurons, dis-organized diencephalic dopaminergic neurons, and the pink1 gene are sensitive markers of oxidative stress in zebrafish	[51,52]
*LRRK2* Morpholino	Motor impairment	Loss of neuronal cells and synuclein aggregation, similar to the phenotype of PD in humans	[53,54,55,56]
*PARK7* Morpholino; *CRISPR*/*Cas9*	Motor impairment	With aging, exhibit lower TH levels, respiratory failure in skeletal muscle, and lower body mass, particularly in the male fish	[57,58,59,60]
*Synuclein* Transgenes	Motor impairment	Led to cell death in larval zebrafish sensory neurons	[61]
*GBA* TALEN	Motor impairment	Reduction of the GBA protein, dopaminergic, and noradrenergic neurons	[62,63]
*PARL* Morpholino; *CRISPR*/*Cas9*	Motor impairment and olfactory dysfunction	Reduced DA neuronal population and dysregulation of the PINK1/Parkin mitophagy pathway	[64,65]

**Table 2 ijms-22-10766-t002:** Zebrafish models of Alzheimer’s disease.

Method	Phenotype	Results	Reference
Amyloid-β42 induced	Intracellular depositions	Link between aging, neurogenesis, regenerative, neuroinflammation, and neural stem cell plasticity	[99,100]
Okadaic acid induced	Cognitive and memory impairments, neuroinflammation cholinergic dysfunction, glutamate excitotoxicity, and mitochondrial dysfunction	Provide all the major molecular hallmarks of AD	[101,102,103]
Cigarette smoke extract induced	Neurocognitive dysfunction	Enhancement of the acetylcholinesterase activity	[104,105]
Aluminum chloride induced	Neurocognitive dysfunction, memory impairment	Impaired locomotor activity, learning, and memory abilities	[106]
Copper induced	Memory impairment	Reduction of the glutathione S-transferase (GST) activity in the gill	[107]
MnCl_2_ induced	Cognition and exploratory behavior	Impairment of aversive long-term memory and distance traveled movement time	[108]
*MAPT* Transgenes	Motor impairment	The phenotypic abnormalities at larval stages make it suitable for high-throughput screening	[109,110]
*PSEN1* ENU-mutagenized	Motor impairment	Regulation of histaminergic neuron development	[111]
*BACE1*/*2* zinc finger nuclease; ENU-mutagenized	Hypomyelination, supernumery neuromasts, and abnormal pigmentation	Bace1 and Bace2 are proteases with different physiological functions	[112]

## Data Availability

Not applicable.

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
