# Peer review of "Zebrafish and Medaka: Important Animal Models for Human Neurodegenerative Diseases"

_ijms, 2021, doi:10.3390/ijms221910766_

Round 1
Reviewer 1 Report
In this manuscript, the authors review some use of zebrafish or medaka as models for various neurodegenerative diseases, mainly Parkinson’s and Alzheimer’s diseases. It is clear that zebrafish has emerged as a new animal model in the recent period in these various diseases which affect the central nervous system with aging. Comprehensive reviews on the interest of zebrafish or medaka are helpful for the numerous researchers who work on these various diseases. However, it is a challenge to write an exhaustive review of the numerous studies in this field in the last years. Accordingly, the present review only addresses some aspect of the abundant literature. Second, although Parkinson and Alzheimer’s disease constitute critical health concerns worldwide, the understanding of their complex mechanisms is still only partial. The critical questions are 1- how the use of zebrafish or medaka can contribute to the understanding of these pathological mechanisms 2- what is the additive value of these fish models compared to the numerous other animal models such as transgenic mice, drosophila and C. elegans? For example, one of the critical question regarding Alzheimer’s disease is how cerebral accumulations of Aβ peptide are induced in sporadic cases which represent a large majority of the affected patients? Is it due to alterations of Aβ transportations from brain to blood or CSF or reduction of Aβintracerebral degradation or both? What is the causes and favoring factors in aging? The authors do not evoke theses aspects. In addition, It is difficult to consider administration of metals such as copper and aluminum as AD relevant models but rather as neurotoxic disorders. I would suggest that the authors focus on one or two of these neurodegenerative diseases and show how the use of zebrafish can contribute to giving enlightening answers on the pathophysiological mechanisms, more effectively than the previously used animal models.
Author Response
Q1: How the use of zebrafish or medaka can contribute to the understanding of these pathological mechanisms? What is the additive value of these fish models compared to the numerous other animal models such as transgenic mice, drosophila and C. elegans?
A: Many thanks. According to the suggestion, we provided the related contents in the manuscript. Please see the “1. Introduction”: “In particular…especially in studies of neurodegenerative diseases”. In addition, we deleted the information about the medaka as advised by another reviewer.
Q2: One of the critical questions regarding Alzheimer’s disease is how cerebral accumulations of Aβ peptide are induced in sporadic cases which represent a large majority of the affected patients? Is it due to alterations of Aβ transportations from brain to blood or CSF or reduction of Aβ intracerebral degradation or both?
A: Thanks for the suggestion. It’s a good concern. According to the suggestion, we added related information in the manuscript. Please see the “Section 3.1”: “Besides, several previous studies…with increased likelihood of plaque formation”.
Q3: What are the causes and favoring factors in aging?
A: Many thanks. According to the suggestion, we added related contents. Please see the “Section 3.1”: “Some previous studies already…Aβ42-induced neurodegeneration”. Thanks for the suggestion.
Q4: In addition, it is difficult to consider administration of metals such as copper and aluminum as AD relevant models but rather as neurotoxic disorders.
A: Thanks for the suggestion. It’s a good concern. We revised the title of “Section 3.2” as advised.
Reviewer 2 Report
Wang and Cao summarise the recent progress in modeling human neurodegenerative diseases in zebrafish and medaka with a particular focus on Parkinson’s, Alzheimer’s and Huntington’s disease. The review is comprehensive and well structured.
I have a few comments and suggestions to be considered. Of the 137 cited publications in this review, I believe that only 2 are experimental papers using medakafish if I am not mistaken. While I am very much in favor to not only consider zebrafish as the one and only teleost model, medakafish has not been very popular so far for ND research. There are reasons for it (for instance the hard chorion making early drug screens very difficult, lack of resources, etc) and there are also reasons why more researchers should actually consider using the medakafish for targeting certain questions – I am just mentioning consistency through highly inbred strains, sex determination genes, etc.. In what way do these two teleosts complement each other? These aspects (and more) could be included by the authors to also justify that “medaka” is in the title of this review.
Other points:
- It would be helpful for non-expert readers to describe briefly the workflow of a drug screen in fish (96-well, small size, transgenic readouts etc)
- Please discuss how early developmental ND phenotypes in fish resemble (or not) late onset human NDs, for instance when using a morpholino approach. What can we learn from this and where are the limits?
- Line 42: The citations “7-9” only resemble zebrafish papers, but the authors refer to zebrafish and medaka. Similarly, in lines 63, 64 the authors mention zebrafish and medaka in connection with Table 1, which however has only zebrafish in the title. This is wrong as reference 59 in this table refers to a medaka paper. There may be other examples – please check.
- Lines 226 – 229: “Furthermore, compared with some mammalian AD models, the advantages of clearly visible forms of behaviour in zebrafish make it a very suitable tool in the cognitive neuroscience analysis [89].” This sentence implies that zebrafish behaviour is easier or better to study when modelling AD than that of mammals. If this is indeed what the authors intend to emphasize, they would need some more justification and/or examples for this statement. If this is not the case, please consider re-phrasing.
- Line 372 - 374: “Conversely, Yeh et al. constructed two transient loss-of-function zebrafish lines (C9orf72u-DENN, C9orf72c-DENN) usinga morpholino injection.”To my knowledge it is not possible to generate fish lines with morpholinos. Yeh et al., describe a transient knock down. Please check and re-phrase.
Author Response
Q1: I have a few comments and suggestions to be considered. Of the 137 cited publications in this review, I believe that only 2 are experimental papers using medakafish if I am not mistaken. While I am very much in favor to not only consider zebrafish as the one and only teleost model, medaka fish has not been very popular so far for ND research. There are reasons for it (for instance the hard chorion making early drug screens very difficult, lack of resources, etc) and there are also reasons why more researchers should actually consider using the medaka fish for targeting certain questions – I am just mentioning consistency through highly inbred strains, sex determination genes, etc. In what way do these two teleosts complement each other? These aspects (and more) could be included by the authors to also justify that “medaka” is in the title of this review.
A: Many thanks! It’s a good concern. According to the suggestion, we deleted the information about the medaka as advised.
Q2: It would be helpful for non-expert readers to describe briefly the workflow of a drug screen in fish (96-well, small size, transgenic readouts etc)
A: Many thanks. We added these contents as advised. Please see the “1. Introduction”: “Finally, drugs can be…high-throughput drug screening scans”.
Q3: Please discuss how early developmental ND phenotypes in fish resemble (or not) late onset human NDs, for instance when using a morpholino approach. What can we learn from this and where are the limits?
A: Thanks for the suggestion. It’s a good concern. Please see the “Section 2.2.1”: “In addition, although neurodegenerative disease…and inability to obtain stable genetic lines”. Thanks for the suggestion.
Q4: Line 42: The citations “7-9” only resemble zebrafish papers, but the authors refer to zebrafish and medaka. Similarly, in lines 63, 64 the authors mention zebrafish and medaka in connection with Table 1, which however has only zebrafish in the title. This is wrong as reference 59 in this table refers to a medaka paper. There may be other examples – please check.
A: We highly appreciate reviewer’s carefulness. As point out by the reviewer before, the medaka papers are not so popular and we deleted the contents about the medaka in the manuscript.
Q5: Lines 226 – 229: “Furthermore, compared with some mammalian AD models, the advantages of clearly visible forms of behaviour in zebrafish make it a very suitable tool in the cognitive neuroscience analysis [89].” This sentence implies that zebrafish behaviour is easier or better to study when modelling AD than that of mammals. If this is indeed what the authors intend to emphasize, they would need some more justification and/or examples for this statement. If this is not the case, please consider re-phrasing.
A: Thanks for the suggestion. We checked the original words from the cited article, and found this description is not correct at all. Therefore, we deleted this sentence.
Q6: Line 372 - 374: “Conversely, Yeh et al. constructed two transient loss-of-function zebrafish lines (C9orf72u-DENN, C9orf72c-DENN) using a morpholino injection.” To my knowledge it is not possible to generate fish lines with morpholinos. Yeh et al., describe a transient knock down. Please check and re-phrase.
A: We highly appreciate reviewer’s carefulness. It’s a good concern. We replaced the “lines” by the “larvae”. Thank you spotting this mistake.
Round 2
Reviewer 2 Report
Wang and Cao have addressed my concerns. However, instead of elaborating the aspect of the (dis-)advantages of medaka and possibly other teleost models, the authors chose to delete this aspect entirely. They state: “According to the suggestion, we deleted the information about the medaka as advised.“ This is certainly not what I intended. Indeed, with the many past and recent reviews on zebrafish in the study of neurodegenerative diseases published, this review has lost what would have made it different to others. I would urge the authors to include at least a separate section on this aspect as implied previously.
Author Response
Q1: Wang and Cao have addressed my concerns. However, instead of elaborating the aspect of the (dis-)advantages of medaka and possibly other teleost models, the authors chose to delete this aspect entirely. They state: “According to the suggestion, we deleted the information about the medaka as advised. “This is certainly not what I intended. Indeed, with the many past and recent reviews on zebrafish in the study of neurodegenerative diseases published, this review has lost what would have made it different to others. I would urge the authors to include at least a separate section on this aspect as implied previously.
A: Thanks for the point of our error. We carefully check the suggestions the reviewer mentioned in the first round. As point out by the reviewer before, we added the contents about the advantages of medaka in the experimental manipulation. Would you kindly please check the “1. Introduction”: “Although the zebrafish is the most…traits and quantitative trait loci”. In addition, we put back into the manuscript the deleted content about the medaka (it was present in the first version of the manuscript submitted). We highly appreciate reviewer's constructive and insightful comments.